# Effect and Mechanism of PINK1/Parkin-Mediated Mitochondrial Autophagy in Rat Lung Injury Induced by Nano Lanthanum Oxide

**DOI:** 10.3390/nano12152594

**Published:** 2022-07-28

**Authors:** Chunyu Chen, Chenxi Zhou, Wenli Zhang, Haiping Liu, Mengfei Wang, Feng Li, Qingzhao Li, Yanhua Cao

**Affiliations:** 1School of Public Health, North China University of Science and Technology, Tangshan 063200, China; 15297688702@163.com (C.C.); liuhaiping202106@163.com (H.L.); wmf1838318@163.com (M.W.); lifeng_litchi@163.com (F.L.); liqingzhao_6@163.com (Q.L.); 2Lin Yi Center for Disease Control and Prevention, Linyi 276100, China; zcx15910016631@163.com; 3Comprehensive Testing and Analyzing Center, North China University of Science and Technology, Tangshan 063200, China; dianj@163.com

**Keywords:** nanometer lanthanum oxide, pulmonary, mitochondrial autophagy injury, PINK1/parkin

## Abstract

Nano lanthanum oxide particles (La_2_O_3_ NPs) are important nanoparticle materials which are widely used in photoelectric production, but their potential health hazards to the respiratory system are not clear. The purpose of this study was to explore the possible mechanism of lung injury induced by La_2_O_3_ NPs. In this study, 40 SPF male SD rats were randomly divided into low-, medium-, and high-dose groups and control groups, with 10 animals in each group. Rats were poisoned by tracheal injection. The low-, medium-, and high-dose groups were given La_2_O_3_ NPs suspension of 25, 50, and 100 mg/kg, respectively, and the control group was given an equal volume of high-temperature sterilized ultrapure water. The rats in each group were exposed once a week for 12 consecutive times. The gene transcription and protein expression levels of PINK1 and parkin in rat lung tissue were mainly detected. Compared with the control group, the gene transcription and protein expression levels of PINK1 and Parkin in the exposed group were significantly higher (*p* < 0.05). La_2_O_3_ NPs may activate PINK1/parkin-induced mitochondrial autophagy.

## 1. Introduction

China is a large country of rare earth elements. Lanthanum is the most widely used rare earth element, and it is widely used in agriculture, animal husbandry, aquaculture, medicine and health, and other industries [1,2]. It can enter and accumulate in the human body through the digestive tract, respiratory tract, placental blood circulation, and breast milk. Lanthanum oxide nanoparticles (La_2_O_3_ NPs) is an important rare earth oxide which has the advantages of high optical activity, high catalytic activity, and strong adsorption selectivity [3]. These characteristics make it good for applications in sensor materials, battery electrode materials, photocatalytic materials, luminescent materials, and other fields, in addition to the field of hydrogen storage materials and magnetic materials.

Due to their small particle size and high biological activity, La_2_O_3_ NPs can easily enter the body through the respiratory tract in the air, accumulate in the lungs, and can also be transported to various tissues and organs with blood circulation, resulting in body damage [4,5]. Shin et al., explored the potential toxicity of La_2_O_3_ NPs after repeated inhalation exposure in male rats and found that La_2_O_3_ NPs could accumulate in rat lung tissue in a dose-dependent manner. Pulmonary inflammation occurred in the exposure group, and after exposure, pulmonary inflammation tended to deteriorate [6]. Lim found that the toxicity of La_2_O_3_ NPs with different particle sizes was similar, but La_2_O_3_ NPs with smaller particle sizes were more toxic to macrophages and alveolar basal epithelial cells, which were absorbed more by the lungs and cleared more slowly. La_2_O_3_ NPs caused more severe lung injury than micron La_2_O_3_ [7]. Sisler et al., found, in their study on the effects of atomized inhalation of La_2_O_3_ NPs on the lungs of mice, that La_2_O_3_ NPs remained in the lungs 56 days after exposure, resulting in a large number of inflammatory cell infiltration in lung tissues, resulting in chronic inflammatory changes and pulmonary fibrosis [8]. Li et al., found, in their study on the gene toxicity of nano-TiO_2_ to mouse lung tissue that nano-TiO_2_ damaged DNA and cells by inducing lung oxidative stress, inflammatory response, and apoptosis [9]. Therefore, La_2_O_3_ NPs can enter the blood circulation and lymphatic circulation through a variety of ways to reach potential target sites, so as to exert toxic effects on the body [10].

Oxidative stress is one of the important mechanisms for nanomaterials to exert their biological toxic effects. Mitochondria are not only the main place for cells to produce ROS, but also the primary target of ROS attack, and a large amount of ROS accumulation will induce mitochondrial autophagy [11]. Mitophagy is a process in which cells selectively clear lesions or damage mitochondria through the mechanism of autophagy [12]. It is one of the key links in the regulation of mitochondrial quality and quantity. Mitochondrial autophagy mediated by the PINK1/parkin pathway is a main mechanism of cell stress regulation induced by nanomaterials [13]. When mitochondria are damaged by oxidative stress, mitochondrial transmembrane potential is destroyed, resulting in the blocking of PINK1 translocation and degradation; this makes PINK1 accumulate on the outer membrane surface of mitochondria, recruits Parkin to damaged mitochondria, and starts mitochondrial autophagy by mediating the activation of Parkin and ubiquitin phosphorylation. When mitochondrial autophagy cannot clear damaged mitochondria or mitochondrial autophagy is excessive, apoptosis-related genes are activated to induce apoptosis [14], resulting in the damage of cells and organs. The participation of mitochondrial autophagy can be seen in the cellular effects of nano materials, both in normal cells and tumor cells. For example, TiO_2_ NPs induce endoplasmic reticulum stress in human trophoblast cells, activate autophagy (including mitochondrial autophagy), and affect placental function [15]; ZnO NPs can kill tumor cells by activating pink1/parkin-mediated mitochondrial autophagy [16].

At present, La_2_O_3_ NPs are being more and more widely used, and their impact on human health and safety has attracted more and more attention. However, the study of mitochondrial autophagy mediated by the PINK1/parkin pathway in lung injury induced by La_2_O_3_ NPs has not been reported. Therefore, this study used the method of the whole animal experiment to establish a rat model of subchronic exposure to La_2_O_3_ NPs through the respiratory tract and to explore the role and specific mechanism of PINK1/parkin signal transduction pathway in lung injury induced by La_2_O_3_ NPs.

## 2. Materials and Methods

### 2.1. Establishment of Animal Model

Forty SPF male SD rats (provided by the Experimental Animal Center of North China University of Science and Technology) weighing 180~220 g were fed freely, drank water at 20~25 °C, and were kept in an environment where the humidity was 45%. They were randomly divided into a control group, La_2_O_3_ NPs low-dose group (25 mg/kg), La_2_O_3_ NPs medium-dose group (50 mg/kg), and La_2_O_3_NPs high-dose group (100 mg/kg), with 10 animals in each group. The test was conducted after 1 week of adaptive feeding.

The appropriate amount of La_2_O_3_ NPs was accurately weighed; sterilized at a high temperature and high pressure; and placed in normal saline to prepare 2.5, 5, and 10 mg/mL La_2_O_3_ NPs suspension. Before the experiment, the mixture was mixed by ultrasonic oscillation for 30 s.

The La_2_O_3_ NPs suspension after ultrasonic oscillation was mixed. After ether anesthesia, the rats were exposed to non-exposed tracheal injection. The low-, medium-, and high-dose groups were given 25, 50, and 100 mg/kg La_2_O_3_ NPs suspension, respectively. The control group was given an equal volume of normal saline (1 mL/100 g). Exposure was once a week and continued for three months, totaling of 12 exposures. The body weight, appearance, eating, and drinking water of the experimental animals were observed and recorded every day.

Rats were sacrificed 24 h after the last exposure, weighed, and recorded. After the heart blood was taken, the lung tissue was quickly removed, and the weight of the tissue was recorded. Some tissues were used for pathological examination, and some blood was separated from serum. The samples were frozen at −80 °C for further use.

### 2.2. General Observation of Rats

During the experiment, the rats’ appearance, activity ability, mental state, food and water intake, and fur changes were observed once a week. Weight was observed every day, with weekly summary analyses of weight changes and timely records.

### 2.3. Determination of Lung Coefficient

After the blood was taken from the heart of the rats, the lung tissue was quickly taken out, rinsed with normal saline, and dried with excessive moisture. Then the wet weight of the lung tissue was accurately weighed with an electronic balance, and the lung coefficient was calculated according to the following formula:
Lung coefficient = (wet lung weight/body weight) × 100%

### 2.4. Determination of Lanthanum Content in Lung Tissue

A total of 1.0 g of rat lung tissue was taken and placed in digestion flasks, respectively, and 4 mL of nitric acid was added overnight until the sample was nitrified. The sample was then evaporated on an electrothermal plate for 1–2 h, until the white powder remained. After cooling, the sample was diluted with nitric acid (1%) to 10 mL, and then a 7500 inductively coupled plasma–atomic emission spectrometry (ICP–MS) analyzer (Santa Clara Agilent) was used to detect the content of lanthanum in lung tissue.

### 2.5. Detection of Oxidative Damage Index in Lung Tissue

The lung tissue was taken with normal saline to prepare 10% tissue homogenate and centrifuged at 3500–4000 r/min for 10 min. The supernatant was taken for use. The activity of SOD and GSH-Px and the content of MDA in lung homogenate were measured by using a kit.

### 2.6. Determination of PINK1 and Parkin mRNA Expression Levels and Protein Expression Levels in Lung

Western blot was used to detect. Take the lung tissue about 0.1 g, cut into pieces, and add 1000 μL lysis liquid. Then use an ultrasonic cell broken instrument for lysis. Protein concentration was measured by using a BCA protein detection kit. The 15 μg protein was subjected to SDS–PAGE electrophoresis, transferred to PVDF membrane, and then blocked with 5% BSA. PVDF membrane was incubated overnight with diluted rabbit anti-mouse PINK1 primary antibody, mouse anti-mouse Parkin primary antibody, and beta-actin. The membrane was washed three times with TBST and incubated with secondary antibody. Finally, ECL imaging was used. The Image J image analysis system was used to analyze the integrated density value (IDV) of protein bands, and the expression levels of PINK1 and Parkin in the lung were calculated with β-actin as the internal reference.

### 2.7. Determination of Bcl-2 and Bax Protein Expression Levels in Lung

The lung tissue was dewaxed and hydrated by immunohistochemical staining and incubated with 1% hydrogen peroxide methanol solution at room temperature for 10 min to inactivate endogenous peroxidase. Double steam washing once, 0.1 mol/L PBS washing 5 min × 3 times; microwave radiation in a microwave oven for 10 min; after the repair solution dropped to room temperature, 0.1 mol/L PBS was washed 5 min × 3 times; normal goat serum blocking solution was added, incubated at room temperature for 20 min; primary antibody (rabbit IgG) was added, left overnight, at 4 °C; 0.01 mmol/L phosphate buffer saline (PBS) was washed 2 min × 3 times; secondary antibody was added, incubated at 37 °C for 20 min; and 0.1 mol/L PBS was washed 5 min × 3 times. DAB color, full washing, dehydration transparent sealing, and image processing and analysis were used.

### 2.8. Pathomorphological Examination of Lung

The middle lobe of the right lung of rats was taken, and the lung tissue with a size of about 1 cm^3^ was retained and fixed with 10% neutral formalin. The lung tissue was routinely dehydrated and embedded in paraffin to prepare paraffin sections. After a 3 mm slice was stained with HE, the morphological changes of rat lungs were observed under light microscope.

### 2.9. Statistical Analysis

The data obtained were expressed as mean ± standard deviation. Excel was used to establish the database, SPSS 17.0 statistical software was used for statistical analysis, one-way analysis of variance (ANOVA) was used to test the significance of the difference between the two groups, and LSD was used to test the comparison between the two groups. A *p* < 0.05 indicated that the difference was statistically significant.

## 3. Results

### 3.1. Characterization of La_2_O_3_ NPs

The size and surface morphology of La_2_O_3_ NPs were examined by using SEM. The SEM images are given in Figure 1A,B, and show that La_2_O_3_ NPs powder has good crystallinity; the product is regular spherical, has good dispersibility, and has uniform particle size distribution. Nanoparticles with an average diameter of 50 nm can be seen. It can be seen from the XRD spectrum that La_2_O_3_ NPs have obvious characteristic diffraction peaks and no other impurity peaks, thus indicating that the La_2_O_3_ NPs have high purity (Figure 1C).

### 3.2. General Condition and Weight Change of Rats

The rats in the control group had shiny fur, quick movement, a sensitive reaction, and good drinking-water intake. The weight of the rats is shown in Figure 2. In the high-dose group, the hair was dry, yellow, and dull, and there was a depilation phenomenon. Autonomous activities were generally reduced, the reaction was slow, and the food intake and water intake were reduced.

The body weight of the high-dose group was significantly higher than that of the control group and the low-dose and medium-dose groups (*p* < 0.05).

### 3.3. Results of Lung Coefficient

Figure 3 showed that lung coefficients of rats in the medium-dose and high-dose groups were significantly higher than those in the control group (*p* < 0.05).

### 3.4. Results of Determination of Lanthanum Content in Lungs

As shown in Figure 4, compared with the control group, the content of lanthanum in the lung tissue of rats in medium- and high-dose groups increased significantly, and the content of lanthanum in the high-dose group was significantly higher than that in the medium-dose group and control group (*p* < 0.05).

### 3.5. Determination Results of SOD, GSH-Px Activities and MDA Content in Lung

From Figure 5, it can be seen that the activities of SOD and GSH-Px in the lungs of the groups with low-, medium-, and high-dose La_2_O_3_ NPs exposure were significantly different from those in the control group (*p* < 0.05). With the increase of exposure dose, the activities of SOD and GSH-Px decreased, the medium-dose group was significantly lower than the low-dose group, and the high-dose group was significantly lower than the medium- and low-dose groups.

Compared with the control group, the content of MDA in the lung of the low-, medium-, and high-dose La_2_O_3_ NPs was significantly different (*p* < 0.05). With the increase of exposure dose, the content of MDA increased, the medium-dose group was significantly higher than the low-dose group, and the high-dose group was significantly higher than the medium- and low-dose groups.

### 3.6. Results of Parkin and PINK1 mRNA Expression Levels in Lung

As shown in Figure 6, the qRT-PCR results showed that PINK1 and Parkin mRNA expression levels increased significantly in the high-dose group relative to the control group (*p* < 0.05).

### 3.7. Results of Parkin and PINK1 Protein Expression Levels in Lung

The Western blotting results are shown in Figure 7. Compared with the control group, the expression levels of total PINK1 and Parkin protein in mitochondria in the high-dose group were significantly higher (*p* < 0.05).

### 3.8. Results of Immunohistochemical

Immunohistochemical staining was employed for characterizing Bax and Bcl-2 on rat lung tissue (Figure 8A,B). Bax-positive cells in the medium- and high-dose groups were significantly increased, and Bcl-2-positive cells in each treatment group were decreased.

### 3.9. Pathological Examination Results of Lung

The pathomorphological changes of rat lung tissue induced by La_2_O_3_ NPs were shown in Figure 9. The control group had normal lung tissue structure, no obvious congestion and edema in alveolar septum, no obvious inflammatory cells in alveolar septum, and uniform thickness of alveolar septum. In the low-dose group, the alveolar size, alveolar septum thickening, and fibroblasts began to increase; in the medium-dose group, inflammatory cell infiltration, alveolar size, alveolar fusion, and fibrous tissue increased; and in the high-dose group, the degree of pulmonary fibrosis was aggravated, and some alveolar cavities disappeared, which was occupied by a large number of collagen fibers, fibroblasts, and lymphocytes.

## 4. Discussion

Nano rare earth materials have a unique surface effect, small size effect, quantum size effect, and macro quantum tunnel effect, which give them good properties and make them more and more widely used; however, their use brings many biosafety problems. La_2_O_3_ NPs can deposit in the trachea and lung tissue through the respiratory tract, and they can also penetrate the cell membrane, causing inflammatory lesions and apoptosis, and eventually leading to lung injury [17,18,19,20,21,22,23,24]. The lung is an important respiratory organ. The research on mitochondrial autophagy in lung injury caused by La_2_O_3_ NPs has not been reported. Therefore, this study established a rat model of subchronic exposure to La_2_O_3_ NPs through the respiratory tract, detected the indicators of oxidative damage in the lung by biochemical methods, and detected the gene transcription and protein expression level of PINK1/parkin by molecular biological methods, so as to explore the effect of La_2_O_3_ NPs exposure on the lung and provide new clues and a theoretical basis for clarifying the toxic mechanism of La_2_O_3_ NPs on the lung.

As a simple and objective quantitative index, body weight can reflect the comprehensive changes of rat health. Feng et al., found that the body weight of offspring in the high-dose LaCl_3_ group was significantly lower than that in the control group, from pregnancy to 5 months of life [25]. The results of this study show that the body weight of rats in the high-dose group is significantly lower than that in the control group (*p* < 0.05), which may be due to the effect of La_2_O_3_ NPs’ exposure on appetite and digestive function, resulting in loss of appetite and weight loss.

In general, the ratio of each organ to body weight is relatively constant. When animals are poisoned, the quality of damaged organs will change, and the organ index will also change. In toxicology, the decline of the organ index means that the organ is atrophic and degenerative; the increase of the organ index may be the hyperemia, edema, and hypertrophic changes of the organ [26]. Sun et al., discussed the chronic pulmonary toxicity caused by continuous intratracheal gavage of TiO_2_ NPs in mice for 90 days. The results showed that TiO_2_ NPs accumulated significantly in the lung, resulting in a significant increase in lung coefficient, inflammation, and pulmonary hemorrhage [27]. Hong et al., found that a large number of lanthanum compounds (LNs) accumulated in the lung, resulting in a significant increase in lung coefficient [28]. The results of this study showed that the lung coefficient of rats in each exposure group was significantly higher than that in the control group (*p* < 0.05), and this may be due to congestion and edema in the lung caused by La_2_O_3_ NPs exposure, which corresponds to the pathological results.

Oxidative stress is one of the important mechanisms of body damage caused by nanoparticles. It refers to the excessive production of highly active molecules such as reactive oxygen species (ROS) in the body when the body is subjected to various harmful stimuli that exceed the scavenging capacity of the body, resulting in the imbalance of oxidation system and antioxidant system, which can lead to tissue damage. The antioxidant system in the organism can effectively remove various endogenous and exogenous free radicals, so as to protect the body. It includes antioxidant enzymes (SOD, CAT, GSH Px, etc.) and antioxidants (GSH, vitamin C, carotene, etc.).

Sod is an important antioxidant enzyme in organisms. It can disproportionate superoxide anion free radicals in tissues and cells to produce hydrogen peroxide, so as to reduce the damage of oxygen free radicals to cells. When the enzyme activity is low, it will cause the accumulation of superoxide anions and generate harmful products such as hydroxyl radicals through other reactions, resulting in lipid peroxidation [29,30]. GSH-Px is an important peroxidase that widely exists in the human body. It can specifically promote peroxide reduction reaction, reduce and block the damage of lipid peroxide, and protect the structural and functional integrity of cell membrane. The activity of SOD and GSH-Px indirectly reflects the ability of the body to scavenge free radicals. MDA is a lipid peroxide formed by reactive oxygen species attacking polyunsaturated fatty acid molecules in biofilm. The higher the content of MDA in the body, the higher the degree of lipid peroxidation. Its content can reflect the degree of oxidative stress injury [31,32,33]. Tian et al., found that lanthanum could accumulate in the liver nuclei and mitochondria of mice, and the content of lanthanum in liver nuclei and mitochondria increased with the increase of exposure dose. At the same time, it was found that lanthanum reduced the activities of various antioxidant enzymes (SOD, CAT, and GPX) in mouse liver nuclei, resulting in oxidative damage of liver nuclei [34]. Li et al., found that lead sulfide nanoparticles can cause SOD activity in rat lung significantly lower than that in the control group, while MDA content significantly increased [35]. Huang et al., studied the potential oxidative damage of lanthanum (LA), cerium (Ce), and neodymium (Nd) on liver nuclei and mitochondria and found that the activities of SOD and CAT decreased in liver nuclei, while the activities of GSH-Px and MDA increased. The activities of SOD, CAT, and GSH-Px in hepatocyte mitochondria decreased significantly, and the level of MDA increased significantly [36]. This study found that the activities of SOD and GSH-Px in the lung of rats exposed to La_2_O_3_ NPs were significantly lower than those in the control group (*p* < 0.05). The content of MDA in the group exposed to La_2_O_3_ NPs was significantly higher than that in the control group (*p* < 0.05). This may be because the accumulation of La_2_O_3_ NPs induces the body to produce a large amount of ROS, resulting in the imbalance of the body’s oxidation and antioxidant system, thus inhibiting the activity of antioxidant enzymes, placing the cells in a state of oxidative stress, and then leading to cell oxidative damage.

When the body is in the state of oxidative stress, excessive ROS accumulated in cells activates mitochondrial autophagy. The PINK1/parkin pathway is the most classical pathway regulating mitochondrial autophagy. PINK1 is a serine/threonine kinase. When mitochondria are damaged by oxidative stress, the mitochondrial transmembrane potential is destroyed, resulting in the blocking of PINK1 translocation and degradation, which makes PINK1 accumulate on the surface of mitochondrial outer membrane and recruit Parkin to damaged mitochondria, so as to start mitochondrial autophagy by mediating the activation of Parkin and ubiquitin phosphorylation [37,38,39]. Parkin is a protein with E3 ubiquitin–protein ligase activity, which mainly mediates substrate ubiquitination and regulates protein degradation and signal transduction. When mitochondria are damaged, E3 ubiquitin ligase activity is activated. After mitochondrial injury, it causes PINK1 accumulation on the outer mitochondrial membrane, mediates the phosphorylation of Parkin and ubiquitin, makes mitochondrial ubiquitin through the positive feedback process, and recruits autophagy receptors to the outer mitochondrial membrane to further form autophagosomes. Autophagosome membrane proteins can specifically recognize and recruit these ubiquitinated proteins into autophagosomes to form complexes with autophagy-related proteins, thus guiding the separation membrane to recognize and encapsulate damaged mitochondria and start mitochondrial autophagy [40,41,42,43,44,45]. Wei et al., found that PINK1/parkin-mediated mitochondrial autophagy was involved in the BV-2 cytotoxicity induced by zinc oxide nanoparticles (ZnO NPs) and established the BV-2 cell model with PINK1 gene knockdown. The study found that ZnO NPs could induce the transport of Parkin from cytoplasm to mitochondria, and the deletion of the PINK1 gene inhibited the recruitment of Parkin to mitochondria, resulting in the failure of cells to trigger mitochondrial autophagy [46]. Wang et al., found that ZnO NPs increased the level of intracellular reactive oxygen species and decreased the mitochondrial membrane potential in a time-dependent manner, which can activate the PINK1/Parkin mediated mitochondrial autophagy process of CAL27 cells [16]. This study found that La_2_O_3_ NPs could significantly increase the gene transcription and protein expression levels of PINK1 and Parkin (*p* < 0.05). The possible reason is that La_2_O_3_ NPs exposure can induce the increase of intracellular ROS level, change the mitochondrial membrane potential, stabilize PINK1 kinase on the mitochondrial outer membrane, recruit and activate Parkin, and trigger mitochondrial autophagy. In the process of mitochondrial autophagy, with the rupture of mitochondrial outer membrane, cytochrome C is released rapidly and induces apoptosis.

The whole process of apoptosis is regulated by the expression of a variety of gene proteins. Bcl-2 family is the key regulator of apoptosis. It is divided into two regulatory proteins, including anti-apoptotic proteins (such as Bcl-2, Bcl-xl, MCL-1, etc.) and pro-apoptotic proteins (such as Bax, Bad, Bcl-xl, Bik, Bid, etc.). Bax inhibits the effect of Bcl-2 and induces apoptosis by forming heterodimer with Bcl-2 [47,48]. It was found that the expression of Bcl-2 decreased in the process of apoptosis [49,50]. Wu et al., found that lanthanum chloride (LaCl_3_) exposure triggered the mitochondrial apoptosis pathway of cortical neurons, upregulated the pro-apoptotic Bax, downregulated the expression of anti-apoptotic Bcl-2, changed the ratio of Bax/Bcl-2, and finally led to neuronal mitochondrial apoptosis [51]. Yang et al., found that LaCl_3_ treatment increased the ratio of pro-apoptotic Bax and anti-apoptotic Bcl-2 proteins, thus breaking the balance between pro-apoptotic and anti-apoptotic Bcl-2 family proteins, promoting the mitochondrial apoptosis pathway, and leading to astrocyte apoptosis [52]. This study found that La_2_O_3_ NPs could significantly increase the expression level of Bax protein in the high-dose group (*p* < 0.05), and the expression level of Bcl-2 protein in the high-dose group was significantly lower than that in the control group. At the same time, the ratio of Bcl-2/Bax in the lung in the group exposed to La_2_O_3_ NPs was significantly lower than that in the control group. This shows that exposure to La_2_O_3_ NPs can cause apoptosis and lead to obvious pathological damage to the lung.

In conclusion, La_2_O_3_ NPs can enter the body through the respiratory tract, accumulate in the lungs, induce the production of a large number of ROS in the lungs, activate the PINK1/parkin-mediated mitochondrial autophagy pathway, promote the occurrence of apoptosis, and finally lead to obvious pathomorphological changes in lung tissue. Due to the complexity of the respiratory system, more research is needed to clarify the exact mechanism of lung tissue injury; moreover, more extensive and in-depth research needs to be conducted on the health hazards and related mechanisms of La_2_O_3_ NPs and lungs.

## Figures and Tables

**Figure 1 nanomaterials-12-02594-f001:**
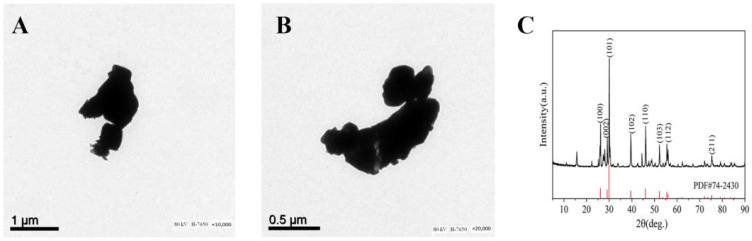
TEM images of La_2_O_3_ NPs ((**A**) 10,000× and (**B**) 20,000×), and XRD pattern (**C**). The black curve represents the La_2_O_3_ NPs sample, and the red curve represents the standard.

**Figure 2 nanomaterials-12-02594-f002:**
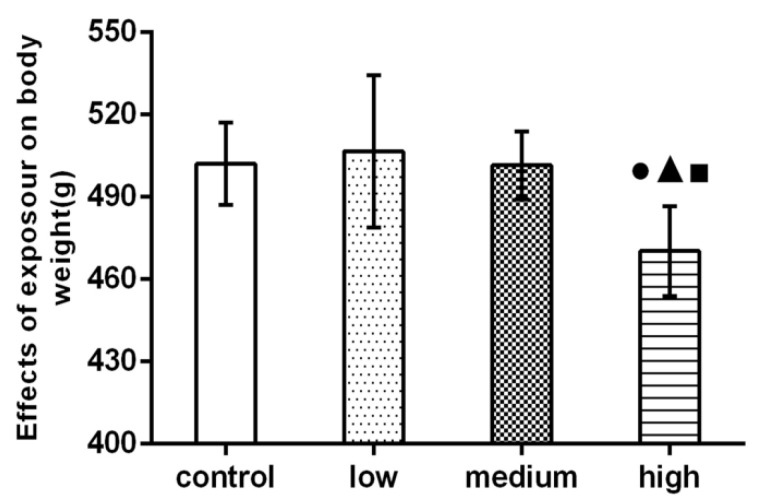
Effect of La_2_O_3_ NPs on rat body weight. Note: ●, compared with the control group, *p* < 0.05; ▲, compared with the low group, *p* < 0.05; ■, compared with the medium group, *p* < 0.05.

**Figure 3 nanomaterials-12-02594-f003:**
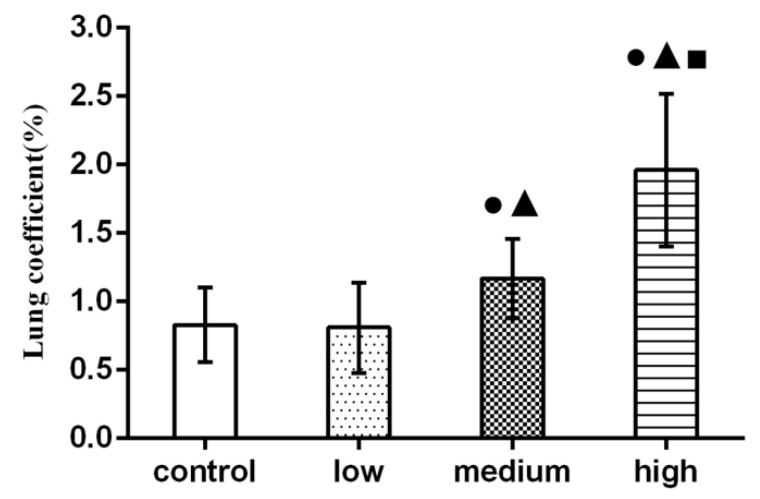
Effect of La_2_O_3_ NPs on lung coefficient in rats. Note: ●, compared with the control group, *p* < 0.05; ▲, compared with the low group, *p* < 0.05; ■, compared with the medium group, *p* < 0.05.

**Figure 4 nanomaterials-12-02594-f004:**
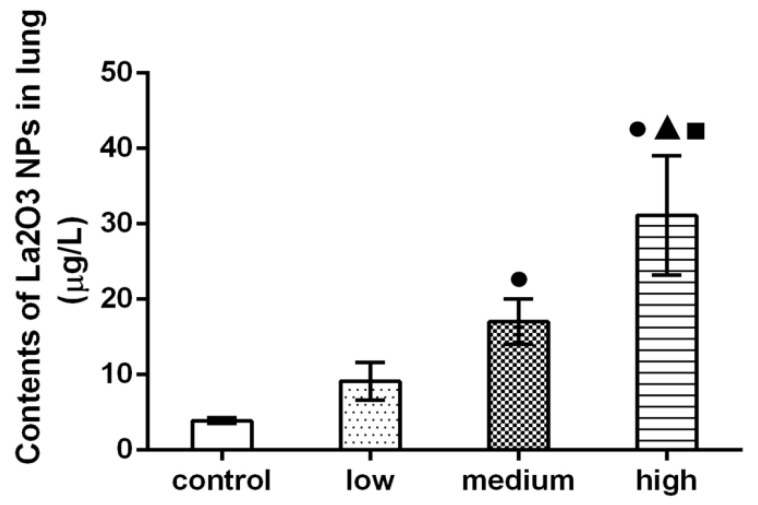
Contents of La_2_O_3_ NPs in lung of rats. Note: ●, compared with the control group, *p* < 0.05; ▲, compared with the low group, *p* < 0.05; ■, compared with the medium group, *p* < 0.05.

**Figure 5 nanomaterials-12-02594-f005:**
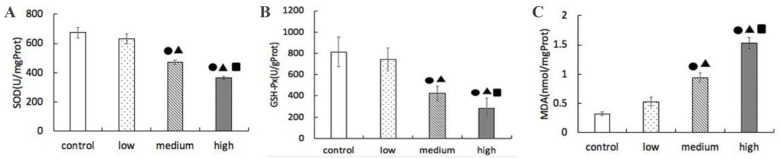
Effects of La_2_O_3_ NPs on SOD, GSH-Px activity, and MDA content in lung tissue of rats in each group. (**A**) SOD level. (**B**) GSH-Px level. (**C**) MDA activity. Note: ●, compared with the control group, *p* < 0.05; ▲, compared with the low group, *p* < 0.05; ■, compared with the medium group, *p* < 0.05.

**Figure 6 nanomaterials-12-02594-f006:**
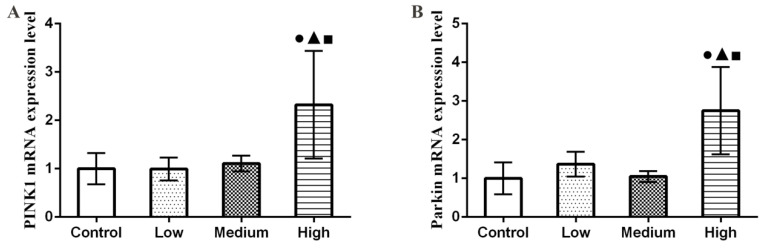
Changes of PINK1 and Parkin mRNA expression levels in rat lung tissue. (**A**) PINK1 mRNA expression level. (**B**) Parkin mRNA expression level. Note: ●, compared with the control group, *p* < 0.05; ▲, compared with the low group, *p* < 0.05; ■, compared with the medium group, *p* < 0.05.

**Figure 7 nanomaterials-12-02594-f007:**
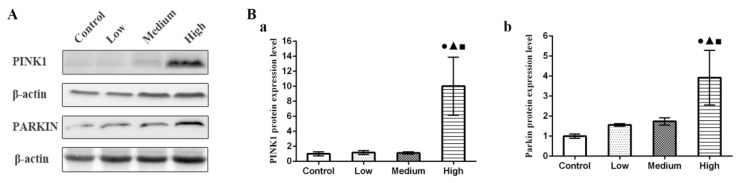
Changes of PINK1 and Parkin protein expression levels in rat lung tissue. (**A**) The expression of PINK1 and parkin proteins in rat lung was detected by Western blot. (**Ba**) PINK1 protein expression level. (**Bb**) Parkin protein expression level. Note: ●, compared with the control group, *p* < 0.05; ▲, compared with the low group, *p* < 0.05; ■, compared with the medium group, *p* < 0.05.

**Figure 8 nanomaterials-12-02594-f008:**
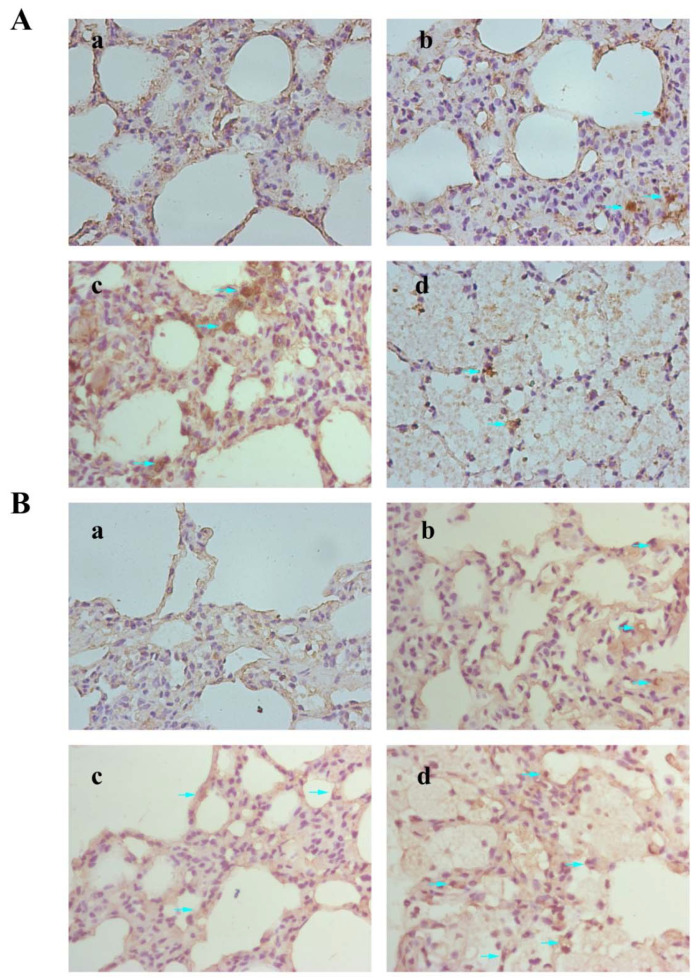
Expressions of Bax (**A**) and Bcl-2 (**B**) in rat lung were detected by immunohistochemical staining. These images were taken at 400× magnification for the control group (**a**), low-dose group (**b**), medium-dose group, (**c**) and high-dose group (**d**). Arrows indicated the positive cells.

**Figure 9 nanomaterials-12-02594-f009:**
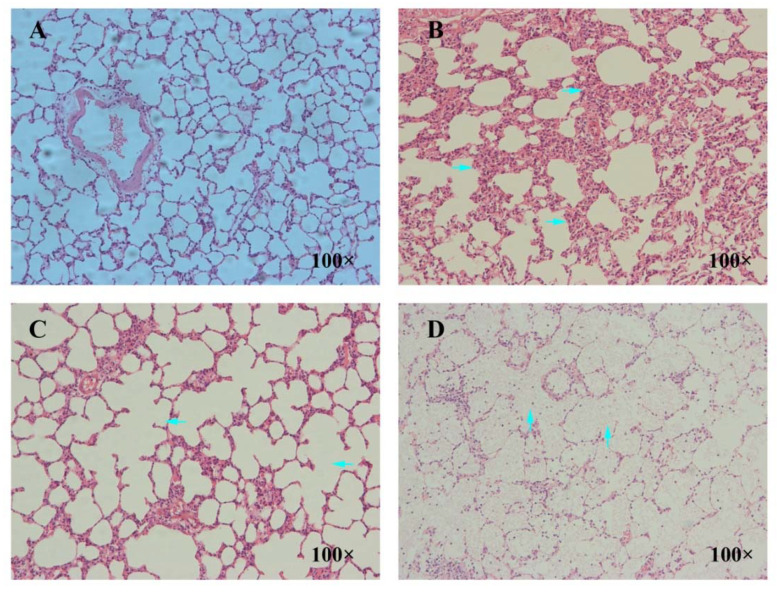
Effect of La_2_O_3_ NPs on pathological morphology of lung tissue in rats. These images were taken at 100× magnification for the control group (**A**), low-dose group (**B**), medium-dose group, (**C**) and high-dose group (**D**). By contrast, the alveolar septum in the control group was thickened (**right** arrows in (**B**)), and alveolar fusion occurred in the medium-dose group (**left** arrows in (**C**)). The alveolar cavity in the high-dose group was filled with collagen fibers, fibroblasts, and lymphocytes (**up** arrows in (**D**)).

## Data Availability

The data are available upon reasonable request from the corresponding author.

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
