# Peer review of "Effect and Mechanism of PINK1/Parkin-Mediated Mitochondrial Autophagy in Rat Lung Injury Induced by Nano Lanthanum Oxide"

_nanomaterials, 2022, doi:10.3390/nano12152594_

Round 1

Reviewer 1 Report

The manuscript entitled “Effect and Mechanism of PINK1/Parkin-mediated Mitochondrial Autophagy in Rat Lung Injury Induced by Nano Lanthanum Oxide" by  Chen et al. have described the synthesis of  La2O3 nanoparticles and checked the autophagy in  La2O3 induced injury in rat lung. The authors should address the following points to improve the quality of manuscript,

1.     How do the authors confirm the electronic and structural properties of the La2O3nanoparticles?

2. The crystallinity of the nanoparticles has not been explained properly. Please provide an appropriate JCPDS file. 

3.     Fig. 1c.: The authors should check the miller indices corresponding to La 5d.

4.     The TEM images showed irregular structure and hence the influence of the nanoparticles on the rat cell should be explained mechanically. Moreover, the ED spectrum displaying atomic % and elemental mapping should be shown.

5.  Graphical representation displayed for expression levels must be varied with (n=3) variations, with its recovery % variations for Parkin, PINKI levels.

6. The mechanism part should be rewritten with the process of autophagy in this experiment and why Bcl-2 +ve cells were decreased after cell treatment.

7. Please mention the superiority of La over other transition metal oxide nanoparticles in reference to ROS (Reactive oxygen species) generation.

8.     In the discussion part the authors have mentioned previous reports trying to explain the current experimental results. However, the superiority of the current experimental set up has not been discussed. The novelty of this work should be properly explained.

9.     More recent references should be cited with further insight on the context of this study.

Reviewer 2 Report

In the manuscript “Effect and Mechanism of PINK1/Parkin-mediated Mitochondrial Autophagy in Rat Lung Injury Induced by Nano Lanthanum Oxide” the authors aimed to explore the possible mechanism of lung injury induced by La2O3 NPs. They used the method of the whole animal experiment to establish a rat model of subchronic exposure to La2O3 NPs through the respiratory tract and to explore the role and specific mechanism of PINK1/parkin signal transduction pathway.

The manuscript is well written, the reader can easily follow the results and discussion. The topic is of significant relevance, so this manuscript could provide important reference information that are of great interest.

The experimental design appears remarkable and is clearly illustrated. The figures are appropriate, in particular the figures 8 and 9. In addition, the authors have consulted a discrete number of scientific papers (51).

Author Response

Dear Editor,

    Thanks for your kind suggestion.We have tried our best to revise the manuscript according to your kind and construction comments and suggestions. Those comments are all valuable and very helpful for revising and improving our paper , as well as the important guiding very helpful for revising and improving our paper , as well as the important guiding significance to our researches. We have studied comments carefully and have made correction which we hope meet with approval. 
    Special thanks to you for your good comments. 

Round 2

Reviewer 1 Report

The answers seem to be okay.